# Information Theoretic Properties of Markov Random Fields, and their Algorithmic Applications

Linus Hamilton[*]          Frederic Koehler [†]          Ankur Moitra [‡]

## Abstract

Markov random fields are a popular model for high-dimensional probability distributions. Over the years, many mathematical, statistical and algorithmic problems on them have been studied. Until recently, the only known algorithms for provably learning them relied on exhaustive search, correlation decay or various incoherence assumptions. Bresler [4] gave an algorithm for learning general Ising models on bounded degree graphs. His approach was based on a structural result about mutual information in Ising models.

Here we take a more conceptual approach to proving lower bounds on the mutual information. Our proof generalizes well beyond Ising models, to arbitrary Markov random fields with higher order interactions. As an application, we obtain algorithms for learning Markov random fields on bounded degree graphs on $n$ nodes with $r$-order interactions in $n^r$ time and $\log n$ sample complexity. Our algorithms also extend to various partial observation models.

## 1  Introduction

### 1.1  Background

*Markov random fields* are a popular model for defining high-dimensional distributions by using a graph to encode conditional dependencies among a collection of random variables. More precisely, the distribution is described by an undirected graph $G = (V, E)$ where to each of the $n$ nodes $u \in V$ we associate a random variable $X_u$ which takes on one of $k_u$ different states. The crucial property is that the conditional distribution of $X_u$ should only depend on the states of $u$'s neighbors. It turns out that as long as every configuration has positive probability, the distribution can be written as

$$\mathbf{Pr}(a_1, \cdots a_n) = \exp \left( \sum_{\ell=1}^{r} \sum_{i_1 < i_2 < \cdots < i_\ell} \theta^{i_1 \cdots i_\ell}(a_1, \cdots a_n) - C \right) \tag{1}$$

Here $\theta^{i_1 \cdots i_\ell} : [k_{i_1}] \times \ldots \times [k_{i_\ell}] \to \mathbb{R}$ is a function that takes as input the configuration of states on the nodes $i_1, i_2, \cdots i_\ell$ and is assumed to be zero on non-cliques. These functions are referred to as *clique potentials*. In the equation above, $C$ is a constant that ensures the distribution is normalized and is called the log-partition function. Such distributions are also called *Gibbs measures* and arise frequently in statistical physics and have numerous applications in computer vision, computational biology, social networks and signal processing. The *Ising model* corresponds to the special case

---

[*]Massachusetts Institute of Technology. Department of Mathematics. Email: `luh@mit.edu`. This work was supported in part by Hertz Fellowship.

[†]Massachusetts Institute of Technology. Department of Mathematics. Email: `fkoehler@mit.edu`.

[‡]Massachusetts Institute of Technology. Department of Mathematics and the Computer Science and Artificial Intelligence Lab. Email: `moitra@mit.edu`. This work was supported in part by NSF CAREER Award CCF-1453261, NSF Large CCF-1565235, a David and Lucile Packard Fellowship and an Alfred P. Sloan Fellowship.

where every node has two possible states and the only non-zero clique potentials correspond to single nodes or to pairs of nodes.

Over the years, many sorts of mathematical, statistical and algorithmic problems have been studied on Markov random fields. Such models first arose in the context of statistical physics where they were used to model systems of interacting particles and predict temperatures at which phase transitions occur [6]. A rich body of work in mathematical physics aims to rigorously understand such phenomena. It is also natural to seek algorithms for sampling from the Gibbs distribution when given its clique potentials. There is a natural Markov chain to do so, and a number of works have identified a critical temperature (in our model this is a part of the clique potentials) above which the Markov chain mixes rapidly and below which it mixes slowly [14, 15]. Remarkably in some cases these critical temperatures also demarcate where approximate sampling goes from being easy to being computationally hard [19, 20]. Finally, various inference problems on Markov random fields lead to graph partitioning problems such as the metric labelling problem [12].

In this paper, we will be primarily concerned with the *structure learning problem*. Given samples from a Markov random field, our goal is to learn the underlying graph $G$ with high probability. The problem of structure learning was initiated by Chow and Liu [7] who gave an algorithm for learning Markov random fields whose underlying graph is a tree by computing the maximum-weight spanning tree where the weight of each edge is equal to the mutual information of the variables at its endpoints. The running time and sample complexity are on the order of $n^2$ and $\log n$ respectively. Since then, a number of works have sought algorithms for more general families of Markov random fields. There have been generalizations to polytrees [10], hypertrees [21] and tree mixtures [2]. Others works construct the neighborhood by exhaustive search [1, 8, 5], impose certain incoherence conditions [13, 17, 11] or require that there are no long range correlations (e.g. between nodes at large distance in the underlying graph) [3, 5].

In a breakthrough work, Bresler [4] gave a simple greedy algorithm that provably works for any bounded degree Ising model, even if it has long-range correlations. This work used mutual information as its underlying progress measure and for each node it constructed its neighborhood. For a set $S$ of nodes, let $X_S$ denote the random variable representing their joint state. Then the key fact is the following:

**Fact 1.1.** *For any node $u$, for any $S \subseteq V \setminus \{u\}$ that does not contain all of $u$'s neighbors, there is a node $v \neq u$ which has non-negligible conditional mutual information (conditioned on $X_S$) with $u$.*

This fact is simultaneously surprising and not surprising. When $S$ contains all the neighbors of $u$, then $X_u$ has zero conditional mutual information (again conditioned on $X_S$) with any other node because $X_u$ only depends on $X_S$. Conversely shouldn't we expect that if $S$ does not contain the entire neighborhood of $u$, that there is some neighbor that has nonzero conditional mutual information with $u$? The difficulty is that the influence of a neighbor on $u$ can be cancelled out indirectly by the other neighbors of $u$. The key fact above tells us that it is impossible for the influences to all cancel out. But is this fact only true for Ising models or is it an instance of a more general phenomenon that holds over any Markov random field?

## 1.2 Our Techniques

In this work, we give a vast generalization of Bresler's [4] lower bound on the conditional mutual information. We prove that it holds in general Markov random fields with higher order interactions provided that we look at sets of nodes. More precisely we prove, in a Markov random field with non-binary states and order up to $r$ interactions, the following fundamental fact:

**Fact 1.2.** *For any node $u$, for any $S \subseteq V \setminus \{u\}$ that does not contain all of $u$'s neighbors, there is a set $I$ of at most $r - 1$ nodes which does not contain $u$ where $X_u$ and $X_I$ have non-negligible conditional mutual information (conditioned on $X_S$).*

Our approach goes through a two-player game that we call the GUESSINGGAME between Alice and Bob. Alice samples a configuration $X_1, X_2, \ldots X_n$ and reveals $I$ and $X_I$ for a randomly chosen set of $u$'s neighbors with $|I| \leq r - 1$. Bob's goal is to guess $X_u$ with non-trivial advantage over its marginal distribution. We give an explicit strategy for Bob that achieves positive expected value. Our approach is quite general because we base Bob's guess on the contribution of $X_I$ to the overall clique potentials that $X_u$ participates in, in a way that the expectation over $I$ yields an unbiased

estimator of the total clique potential. The fact that the strategy has positive expected value is then immediate, and all that remains is to prove a quantitative lower bound on it using the law of total variance. From here, the intuition is that if the mutual information $I(X_u; X_I)$ were zero for all sets $I$ then Bob could not have positive expected value in the GUESSINGGAME.

## 1.3 Our Results

Let $\Gamma(u)$ denote the neighbors of $u$. We require certain conditions (Definition 2.3) on the clique potentials to hold, which we call $\alpha, \beta$-non-degeneracy, to ensure that the presence or absence of each hyperedge can be information-theoretically determined from few samples (essentially that no clique potential is too large and no non-zero clique potential is too small). Under this condition, we prove:

**Theorem 1.3.** *Fix any node $u$ in an $\alpha, \beta$-non-degenerate Markov random field of bounded degree and a subset of the vertices $S$ which does not contain the entire neighborhood of $u$. Then taking $I$ uniformly at random from the subsets of the neighbors of $u$ not contained in $S$ of size $s = \min(r - 1, |\Gamma(u) \setminus S|)$, we have $\mathbf{E}_I[I(X_u; X_I | X_S)] \geq C$.*

See Theorem 4.3 which gives the precise dependence of $C$ on all of the constants, including $\alpha$, $\beta$, the maximum degree $D$, the order of the interactions $r$ and the upper bound $K$ on the number of states of each node. We remark that $C$ is exponentially small in $D$, $r$ and $\beta$ and there are examples where this dependence is necessary [18].

Next we apply our structural result within Bresler's [4] greedy framework for structure learning to obtain our main algorithmic result. We obtain an algorithm for learning Markov random fields on bounded degree graphs with a logarithmic number of samples, which is information-theoretically optimal [18]. More precisely we prove:

**Theorem 1.4.** *Fix any $\alpha, \beta$-non-degenerate Markov random field on $n$ nodes with $r$-order interactions and bounded degree. There is an algorithm for learning $G$ that succeeds with high probability given $C' \log n$ samples and runs in time polynomial in $n^r$.*

**Remark 1.5.** It is easy to encode an $r - 1$-sparse parity with noise as a Markov random field with order $r$ interactions. This means if we could improve the running time to $n^{o(r)}$ this would yield the first $n^{o(k)}$ algorithm for learning $k$-sparse parities with noise, which is a long-standing open question. The best known algorithm of Valiant [22] runs in time $n^{0.8k}$.

See Theorem 5.1 for a more precise statement. The constant $C'$ depends doubly exponentially on $D$. In the special case of Ising models with no external field, Vuffray et al. [23] gave an algorithm based on convex programming that reduces the dependence on $D$ to singly exponential. In greedy approaches based on mutual information like the one we consider here, doubly-exponential dependence on $D$ seems intrinsic. As in Bresler's [4] work, we construct a superset of the neighborhood that contains roughly $1/C$ nodes where $C$ comes from Theorem 1.3. Recall that $C$ is exponentially small in $D$. Then to accurately estimate conditional mutual information when conditioning on the states of this many nodes, we need doubly exponential in $D$ many samples.

Our results extend to a model where we are only allowed partial observations. More precisely, for each sample we are allowed to specify a set $J$ of size at most $C''$ and all we observe is $X_J$. We prove:

**Theorem 1.6.** *Fix any $\alpha, \beta$-non-degenerate Markov random field on $n$ nodes with $r$-order interactions and bounded degree. There is an algorithm for learning $G$ with $C''$-bounded queries that succeeds with high probability given $C' \log n$ samples and runs in time polynomial in $n^r$.*

See Theorem 5.3 for a more precise statement. This is a natural scenario that arises when it is too expensive to obtain a sample where the states of all nodes are known. We also consider a model where each node's state is erased (and unobserved) independently with some fixed probability $p$. See the supplementary material for a precise statement. The fact that we can straightforwardly obtain algorithms for these alternative settings demonstrates the flexibility of greedy, information-theoretic approaches to learning.

## 2 Preliminaries

For reference, all fundamental parameters of the graphical model (max degree, etc.) are defined in the next two subsections. In terms of these fundamental parameters, we define additional parameters $\gamma$ and $\delta$ in (3), $C'(\gamma, K, \alpha)$ in Theorem 4.3, and $\tau$ in (5) and $L$ in (6).

### 2.1 Markov Random Fields and the Canonical Form

Let $K$ be an upper bound on the maximum number of states of any node. Recall the joint probability distribution of the model, given in (1). For notational convenience, even when $i_1, \ldots, i_\ell$ are not sorted in increasing order, we define $\theta^{i_1 \cdots i_\ell}(a_1, \ldots, a_\ell) = \theta^{i'_1 \cdots i'_\ell}(a'_1, \ldots, a'_\ell)$ where the $i'_1, \ldots, i'_\ell$ are the sorted version of $i_1, \ldots, i_\ell$ and the $a'_1, \ldots, a'_\ell$ are the corresponding copies of $a_1, \ldots, a_\ell$.

The parameterization in (1) is not unique. It will be helpful to put it in a normal form as below. A *tensor fiber* is the vector given by fixing all of the indices of the tensor except for one; this generalizes the notion of row/column in matrices. For example for any $1 \leq m \leq \ell$, $i_1 < \ldots < i_m < \ldots i_\ell$ and $a_1, \ldots, a_{m-1}, a_{m+1}, \ldots a_\ell$ fixed, the corresponding tensor fiber is the set of elements $\theta^{i_1 \cdots i_\ell}(a_1, \ldots, a_m, \ldots, a_\ell)$ where $a_m$ ranges from 1 to $k_{i_m}$.

**Definition 2.1.** We say that the weights $\theta$ are in *canonical form*[4] if for every tensor $\theta^{i_1 \cdots i_\ell}$, the sum over all of the *tensor fibers* of $\theta^{i_1 \cdots i_\ell}$ is zero.

Moreover we say that a tensor with the property that the sum over all tensor fibers is zero is a *centered tensor*. Hence having a Markov random field in canonical form just means that all of the tensors corresponding to its clique potentials are centered. We observe that every Markov random field can be put in canonical form:

**Claim 2.2.** *Every Markov random field can be put in canonical form*

### 2.2 Non-Degeneracy

Let $\mathcal{H} = (V, H)$ denote a hypergraph obtained from the Markov random field as follows. For every non-zero tensor $\theta^{i_1 \cdots i_\ell}$ we associate a hyperedge $(i_1 \cdots i_\ell)$. We say that a hyperedge $h$ is maximal if no other hyperedge of strictly larger size contains $h$. Now $G = (V, E)$ can be obtained by replacing every hyperedge with a clique. Let $D$ be a bound on the maximum degree. Recall that $\Gamma(u)$ denotes the neighbors of $u$. We will require the following conditions in order to ensure that the presence and absence of every maximal hyperedge is information-theoretically determined:

**Definition 2.3.** We say that a Markov random field is $\alpha, \beta$-non-degenerate if

(a) Every edge $(i, j)$ in the graph $G$ is contained in some hyperedge $h \in H$ where the corresponding tensor is non-zero.

(b) Every maximal hyperedge $h \in H$ has at least one entry lower bounded by $\alpha$ in absolute value.

(c) Every entry of $\theta^{i_1 i_2 \cdots i_\ell}$ is upper bounded by a constant $\beta$ in absolute value.

We will refer to a hyperedge $h$ with an entry lower bounded by $\alpha$ in absolute value as $\alpha$-*nonvanishing*.

### 2.3 Bounds on Conditional Probabilities

First we review properties of the conditional probabilities in a Markov random field as well as introduce some convenient notation which we will use later on. Fix a node $u$ and its neighborhood $U = \Gamma(u)$. Then for any $R \in [k_u]$ we have

$$P(X_u = R | X_U) = \frac{\exp(\mathcal{E}^X_{u,R})}{\sum_{B=1}^{k_u} \exp(\mathcal{E}^X_{u,B})} \tag{2}$$

where we define

$$\mathcal{E}_{u,R}^{X} = \sum_{\ell=1}^{r} \sum_{i_2 < \cdots < i_\ell} \theta^{u i_2 \cdots i_\ell}(R, X_{i_2}, \cdots, X_{i_\ell})$$

and $i_2, \ldots, i_\ell$ range over elements of the neighborhood $U$; when $\ell = 1$ the inner sum is just $\theta^u(R)$. Let $X_{\sim u} = X_{[n] \backslash \{u\}}$. To see that the above is true, first condition on $X_{\sim u}$, and observe that the probability for a certain $X_u$ is proportional to $\exp(\mathcal{E}_{u,R}^{X})$, which gives the right hand side of (2). Then apply the tower property for conditional probabilities.

Therefore if we define (where $|T|_{max}$ denotes the maximum entry of a tensor $T$)

$$\gamma := \sup_{u} \sum_{\ell=1}^{r} \sum_{i_2 < \cdots < i_\ell} |\theta^{u i_2 \cdots i_\ell}|_{max} \leq \beta \sum_{\ell=1}^{r} \binom{D}{\ell-1}, \qquad \delta := \frac{1}{K} \exp(-2\gamma) \qquad (3)$$

then for any $R$

$$P(X_u = R | X_U) \geq \frac{\exp(-\gamma)}{K \exp(\gamma)} = \frac{1}{K} \exp(-2\gamma) = \delta \qquad (4)$$

Observe that if we pick any node $i$ and consider the new Markov random field given by conditioning on a fixed value of $X_i$, then the value of $\gamma$ for the new Markov random field is non-increasing.

## 3 The Guessing Game

Here we introduce a game-theoretic framework for understanding mutual information in general Markov random fields. The GUESSINGGAME is defined as follows:

---

1. Alice samples $X = (X_1, \ldots, X_n)$ and $X' = (X'_1, \ldots, X'_n)$ independently from the Markov random field
2. Alice samples $R$ uniformly at random from $[k_u]$
3. Alice samples a set $I$ of size $s = \min(r - 1, d_u)$ uniformly at random from the neighbors of $u$
4. Alice tells Bob $I$, $X_I$ and $R$
5. Bob wagers $w$ with $|w| \leq \gamma K \binom{D}{r-1}$
6. Bob gets $\Delta = w \mathbb{1}_{X_u = R} - w \mathbb{1}_{X'_u = R}$

---

Bob's goal is to guess $X_u$ given knowledge of the states of some of $u$'s neighbors. The Markov random field (including all of its parameters) are common knowledge. The intuition is that if Bob can obtain a positive expected value, then there must be some set $I$ of neighbors of $u$ which have non-zero mutual information. In this section, will show that there is a simple, explicit strategy for Bob that yields positive expected value.

### 3.1 A Good Strategy for Bob

Here we will show an explicit strategy for Bob that has positive expected value. Our analysis will rest on the following key lemma:

**Lemma 3.1.** *There is a strategy for Bob that wagers at most $\gamma K \binom{D}{r-1}$ in absolute value that satisfies*

$$\mathop{\mathbf{E}}_{I, X_I} [w | X_{\sim u}, R] = \mathcal{E}_{u,R}^{X} - \sum_{B \neq R} \mathcal{E}_{u,B}^{X}$$

*Proof.* First we explicitly define Bob's strategy. Let

$$\Phi(R, I, X_I) = \sum_{\ell=1}^{s} C_{u,\ell,s} \sum_{i_1 < i_2 < \cdots < i_\ell} \mathbb{1}_{\{i_1 \cdots i_\ell\} \subseteq I} \theta^{u i_1 \cdots i_\ell}(R, X_{i_1}, \ldots, X_{i_\ell})$$

where $C_{u,\ell,s} = \frac{\binom{d_u}{s}}{\binom{d_u-\ell}{s-\ell}}$. Then Bob wagers

$$w = \Phi(R, I, X_I) - \sum_{B \neq R} \Phi(B, I, X_I)$$

Notice that the strategy only depends on $X_I$ because all terms in the summation where $\{i_1 \cdots i_\ell\}$ are not a subset of $I$ have zero contribution.

The intuition behind this strategy is that the weighting term satisifes

$$C_{u,\ell,s} = \frac{1}{\mathbf{Pr}[\{i_1, \ldots i_\ell\} \subset I]}$$

Thus when we take the expectation over $I$ and $X_I$ we get

$$\mathop{\mathbf{E}}_{I,X_I}[\Phi(R, I, X_I)|X_{\sim u}, R] = \sum_{\ell=1}^{r} \sum_{i_2 < \cdots < i_\ell} \theta^{u i_2 \cdots i_\ell}(R, X_{i_2}, \cdots, X_{i_\ell}) = \mathcal{E}_{u,R}^X$$

and hence $\mathbf{E}_{I,X_I}[w|X_{\sim u}, R] = \mathcal{E}_{u,R}^X - \sum_{B \neq R} \mathcal{E}_{u,B}^X$. To complete the proof, notice that $C_{u,\ell,s} \leq \binom{D}{r-1}$ which using the definition of $\gamma$ implies that $|\Phi(R, I, X_I)| \leq \gamma \binom{D}{r-1}$ for any state $B$, and thus Bob wagers at most the desired amount (in absolute value). $\qquad\square$

Now we are ready to analyze the strategy:

**Theorem 3.2.** *There is a strategy for Bob that wagers at most $\gamma K \binom{D}{r-1}$ in absolute value which satisfies*

$$\mathbf{E}[\Delta] \geq \frac{4\alpha^2 \delta^{r-1}}{r^{2r} e^{2\gamma}}$$

*Proof.* We will use the strategy from Lemma 3.1. First we fix $X_{\sim u}$, $X'_{\sim u}$ and $R$. Then we have

$$\mathop{\mathbf{E}}_{I,X_I}[\Delta|X_{\sim u}, X'_{\sim u}, R] = \mathop{\mathbf{E}}_{I,X_I}[w|X_{\sim u}, R]\Big( \mathbf{Pr}[X_u = R|X_{\sim u}, R] - \mathbf{Pr}[X'_u = R|X'_{\sim u}, R]\Big)$$

which follows because $\Delta = r\mathbb{1}_{X_u=R} - r\mathbb{1}_{X'_u=R}$ and because $r$ and $X_u$ do not depend on $X'_{\sim u}$ and similarly $X'_u$ does not depend on $X_{\sim u}$. Now using (2) we calculate:

$$\mathbf{Pr}[X_u = R|X_{\sim u}, R] - \mathbf{Pr}[X'_u = R|X'_{\sim u}, R] = \frac{\exp(\mathcal{E}_{u,R}^X)}{\sum_B \exp(\mathcal{E}_{u,B}^X)} - \frac{\exp(\mathcal{E}_{u,R}^{X'})}{\sum_B \exp(\mathcal{E}_{u,B}^{X'})}$$

$$= \frac{1}{D}\Big( \sum_{B \neq R} \exp(\mathcal{E}_{u,R}^X + \mathcal{E}_{u,B}^{X'}) - \exp(\mathcal{E}_{u,B}^X + \mathcal{E}_{u,R}^{X'})\Big)$$

where $D = \Big(\sum_B \exp(\mathcal{E}_{u,B}^X)\Big)\Big(\sum_B \exp(\mathcal{E}_{u,B}^{X'})\Big)$. Thus putting it all together we have

$$\mathop{\mathbf{E}}_{I,X_I}[\Delta|X_{\sim u}, X'_{\sim u}, R] = \frac{1}{D}\Big(\mathcal{E}_{u,R}^X - \sum_{B \neq R} \mathcal{E}_{u,B}^X\Big)\Big( \sum_{B \neq R} \exp(\mathcal{E}_{u,R}^X + \mathcal{E}_{u,B}^{X'}) - \exp(\mathcal{E}_{u,B}^X + \mathcal{E}_{u,R}^{X'})\Big)$$

Now it is easy to see that

$$\sum_{\text{distinct } R,G,B} \mathcal{E}_{u,B}^X \left( \sum_{G \neq R} \exp(\mathcal{E}_{u,R}^X + \mathcal{E}_{u,G}^{X'}) - \exp(\mathcal{E}_{u,G}^X + \mathcal{E}_{u,R}^{X'}) \right) = 0$$

which follows because when we interchange $R$ and $G$ the entire term multiplies by a negative one and so we can pair up the terms in the summation so that they exactly cancel. Using this identity we get

$$\mathop{\mathbf{E}}_{I,X_I}[\Delta|X_{\sim u}, X'_{\sim u}] = \frac{1}{k_u D} \sum_R \sum_{B \neq R} \Big(\mathcal{E}_{u,R}^X - \mathcal{E}_{u,B}^X\Big)\Big( \exp(\mathcal{E}_{u,R}^X + \mathcal{E}_{u,B}^{X'}) - \exp(\mathcal{E}_{u,B}^X + \mathcal{E}_{u,R}^{X'})\Big)$$

where we have also used the fact that $R$ is uniform on $k_u$. And finally using the fact that $X_{\sim u}$ and $X'_{\sim u}$ are identically distributed we can sample $Y_{\sim u}$ and $Z_{\sim u}$ and flip a coin to decide whether we set $X_{\sim u} = Y_{\sim u}$ and $X'_{\sim u} = Z_{\sim u}$ or vice-versa. Now we have

$$\mathop{\mathbf{E}}_{I,X_I}[\Delta|Y_{\sim u}, Z_{\sim u}] = \frac{1}{2k_u D} \sum_R \sum_{B \neq R} \left( \mathcal{E}_{u,R}^Y - \mathcal{E}_{u,B}^Y - \mathcal{E}_{u,R}^Z + \mathcal{E}_{u,B}^Z \right)\left( e^{\mathcal{E}_{u,R}^Y + \mathcal{E}_{u,B}^Z} - e^{\mathcal{E}_{u,B}^Y + \mathcal{E}_{u,R}^Z} \right)$$

With the appropriate notation it is easy to see that the above sum is strictly positive. Let $a_{R,B} = \mathcal{E}_{u,R}^Y + \mathcal{E}_{u,B}^Z$ and $b_{R,B} = \mathcal{E}_{u,R}^Z + \mathcal{E}_{u,B}^Y$. With this notation:

$$\mathop{\mathbf{E}}_{I,X_I}[\Delta|Y_{\sim u}, Z_{\sim u}] = \frac{1}{2Dk_u} \sum_R \sum_{B \neq R} \left( a_{R,B} - b_{R,B} \right)\left( \exp(a_{R,B}) - \exp(b_{R,B}) \right)$$

Since $\exp(x)$ is a strictly increasing function it follows that as long as $a_{R,B} \neq b_{R,B}$ for some term in the sum, the sum is positive. In Lemma 3.3 we prove that the expectation over $Y$ and $Z$ of this sum is at least $\frac{4\alpha^2 \delta^{r-1}}{r^{2r} e^{2\gamma}}$, which completes the proof. $\qquad\square$

In the supplementary material we show how to use the law of total variance to give a quantitative lower bound on the sum that arose in the proof of Theorem 3.2. More precisely we show:

**Lemma 3.3.**

$$\mathop{\mathbf{E}}_{Y,Z} \left[ \sum_R \sum_{B \neq R} \left( \mathcal{E}_{u,R}^Y - \mathcal{E}_{u,B}^Y - \mathcal{E}_{u,R}^Z + \mathcal{E}_{u,B}^Z \right)\left( \exp(\mathcal{E}_{u,R}^Y + \mathcal{E}_{u,B}^Z) - \exp(\mathcal{E}_{u,B}^Y + \mathcal{E}_{u,R}^Z) \right) \right] \geq \frac{4\alpha^2 \delta^{r-1}}{r^{2r} e^{2\gamma}}$$

# 4   Implications for Mutual Information

In this section we show that Bob's strategy implies a lower bound on the mutual information between node $u$ and a subset $I$ of its neighbors of size at most $r - 1$. We then extend the argument to work with conditional mutual information as well.

## 4.1   Mutual Information in Markov Random Fields

Recall that the goal of the GUESSINGGAME is for Bob to use information about the states of nodes $I$ to guess the state of node $u$. Intuitively, if $X_I$ conveys no information about $X_u$ then it should contradict the fact that Bob has a strategy with positive expected value. We make this precise below. Our argument proceeds in two steps. First we upper bound the expected value of any strategy.

**Lemma 4.1.** *For any strategy,*

$$\mathbf{E}[\Delta] \leq \gamma K \binom{D}{r-1} \mathop{\mathbf{E}}_{I, X_I, R} \left[ |\mathbf{Pr}[X_u = R|X_I] - \mathbf{Pr}[X_u = R]| \right]$$

Intuitively this follows because Bob's optimal strategy given $I$, $X_I$ and $R$ is to guess

$$w = \text{sgn}(\mathbf{Pr}[X_u = R|X_I] - \mathbf{Pr}[X_u = R])\gamma K$$

Next we lower bound the mutual information using (essentially) the same quantity. We prove

**Lemma 4.2.**

$$\sqrt{\frac{1}{2} I(X_u; X_I)} \geq \frac{1}{K^r} \mathop{\mathbf{E}}_{X_I, R} \left[ |\mathbf{Pr}(X_u = R|X_I) - \mathbf{Pr}(X_u = R)| \right]$$

These bounds together yield a lower bound on the mutual information. In the supplementary material, we show how to extend the lower bound for mutual information to conditional mutual information. The main idea is to show there is a setting of $X_S$ where the hyperedges do not completely cancel out in the Markov random field we obtain by conditioning on $X_S$.

**Theorem 4.3.** *Fix a vertex $u$ such that all of the maximal hyperedges containing $u$ are $\alpha$-nonvanishing, and a subset of the vertices $S$ which does not contain the entire neighborhood of*

$u$. *Then taking $I$ uniformly at random from the subsets of the neighbors of $u$ not contained in $S$ of size $s = \min(r - 1, |\Gamma(u) \setminus S|)$,*

$$\mathop{\mathbf{E}}_{I}\left[\sqrt{\frac{1}{2}I(X_u; X_I|X_S)}\right] \geq C'(\gamma, K, \alpha)$$

*where explicitly*

$$C'(\gamma, K, \alpha) := \frac{4\alpha^2\delta^{r+d-1}}{r^{2r}K^{r+1}\binom{D}{r-1}\gamma e^{2\gamma}}$$

## 5 Applications

We now employ the greedy approach of Bresler [4] which was previously used to learn Ising models on bounded degree graphs. Suppose we are given $m$ independent samples from the Markov random field. Let $\widehat{\mathbf{Pr}}$ denote the empirical distribution and let $\widehat{\mathbf{E}}$ denote the expectation under this distribution.

We compute empirical estimates for a certain information theoretic quantity $\nu_{u,I|S}$ (defined in the supplementary material) as follows

$$\widehat{\nu}_{u,I|S} := \mathop{\mathbf{E}}_{R,G} \widehat{\mathbf{E}}_{X_S}[|\widehat{\mathbf{Pr}}(X_u = R, X_I = G|X_S) - \widehat{\mathbf{Pr}}(X_u = R|X_S)\widehat{\mathbf{Pr}}(X_I = G|X_S)|]$$

where $R$ is a state drawn uniformly at random from $[k_u]$, and $G$ is an $|I|$-tuple of states drawn independently uniformly at random from $[k_{i_1}] \times [k_{i_2}] \times \ldots \times [k_{i_{|I|}}]$ where $I = (i_1, i_2, \ldots i_{|I|})$. Also we define $\tau$ (which will be used as a thresholding constant) as

$$\tau := C'(\gamma, k, \alpha)/2 \tag{5}$$

and $L$, which is an upper bound on the size of the superset of a neighborhood of $u$ that the algorithm will construct,

$$L := (8/\tau^2)\log K = (32/C'(\gamma, k, \alpha)^2)\log K. \tag{6}$$

Then the algorithm MRFNBHD at node $u$ is:

---

1. Fix input vertex $u$. Set $S := \emptyset$.
2. While $|S| \leq L$ and there exists a set of vertices $I \subset [n] \setminus S$ of size at most $r - 1$ such that $\widehat{\nu}_{u,I|S} > \tau$, set $S := S \cup I$.
3. For each $i \in S$, if $\widehat{\nu}_{u,i|S\setminus i} < \tau$ then remove $i$ from $S$.
4. Return set $S$ as our estimate of the neighborhood of $u$.

---

**Theorem 5.1.** *Fix $\omega > 0$. Suppose we are given $m$ samples from an $\alpha, \beta$-non-degenerate Markov random field with $r$-order interactions where the underlying graph has maximum degree at most $D$ and each node takes on at most $K$ states. Suppose that*

$$m \geq \frac{60K^{2L}}{\tau^2\delta^{2L}}\Big(\log(1/\omega) + \log(L + r) + (L + r)\log(nK) + \log 2\Big).$$

*Then with probability at least $1 - \omega$, MRFNBHD when run starting from each node $u$ recovers the correct neighborhood of $u$, and thus recovers the underlying graph $G$. Furthermore, each run of the algorithm takes $O(mLn^r)$ time.*

In many situations, it is too expensive to obtain full samples from a Markov random field (e.g. this could involve needing to measure every potential symptom of a patient). Here we consider a model where we are allowed only partial observations in the form of a $C$-bounded query:

**Definition 5.2.** A $C$-*bounded query* to a Markov random field is specified by a set $S$ with $|S| \leq C$ and we observe $X_S$

Our algorithm MRFNBHD can be made to work with $C$-bounded queries instead of full observations. We prove:

**Theorem 5.3.** *Fix an $\alpha, \beta$-non-degenerate Markov random field with $r$-order interactions where the underlying graph has maximum degree at most $D$ and each node takes on at most $K$ states. The bounded queries modification to the algorithm returns the correct neighborhood of every vertex $u$ using $m'Lrn^r$-bounded queries of size at most $L + r$ where*

$$m' = \frac{60K^{2L}}{\tau^2\delta^{2L}}\Big(\log(Lrn^r/\omega) + \log(L+r) + (L+r)\log(nK) + \log 2\Big),$$

*with probability at least $1 - \omega$.*

In the supplementary material, we extend our results to the setting where we observe partial samples where the state of each node is revealed independently with probability $p$, and the choice of which nodes to reveal is independent of the sample.

**Acknowledgements:** We thank Guy Bresler for valuable discussions and feedback.

## Footnotes

[4]This is the same as writing the log of the probability mass function according to the *Efron-Stein decomposition* with respect to the uniform measure on colors; this decomposition is known to be unique. See e.g. Chapter 8 of [16]

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
