[Supplementary Material]

# Supplementary Material for "Information Theoretic Properties of Markov Random Fields, and their Algorithmic Applications"

## A  Omitted Proofs in Section 2

### A.1  Markov Random Fields and the Canonical Form

**Claim A.1.** *Every Markov random field can be put in canonical form*

*Proof.* We will recenter the tensors one by one without changing the law in (1). Starting with an arbitrary parameterization, observe that if the sum along some tensor fiber is $s \neq 0$, we can subtract $s/k_{i_m}$ from each of the entries in the tensor fiber, so the sum over the tensor fiber is now zero, and add $s$ to $\theta^{i \sim m}(a_{\sim m})$ without changing the law of $X$ in (1). Here $i_{\sim m}$ is our notation for $i_1, \ldots, i_{m-1}, i_{m+1}, \ldots i_\ell$. By iterating this process from the tensors representing the highest-order interactions down to the tensors representing the lowest-order interactions[1], we obtain the desired canonical form. □

### A.2  Lower Bounds for Conditional Mutual Information

As in Bresler's work on learning Ising models [1], certain information theoretic quantities will play a crucial role as a progress measure in our algorithms. Specifically, we will use the functional

$$\nu_{u,I|S} := \mathop{\mathbf{E}}_{R,G}\left[\mathop{\mathbf{E}}_{X_S}\left[\left|\mathbf{Pr}(X_u = R, X_I = G|X_S) - \mathbf{Pr}(X_u = R|X_S)\,\mathbf{Pr}(X_I = G|X_S)\right|\right]\right]$$

where $R$ is a state drawn uniformly at random from $[k_u]$, uniformly at random and $G$ is an $|I|$-tuple of states drawn independently uniformly at random from $[k_{i_1}] \times [k_{i_2}] \times \ldots \times [k_{i_{|I|}}]$ where $I = (i_1, i_2, \ldots i_{|I|})$. This will be used as a *proxy* for conditional mutual information which can be efficiently estimated from samples. The following lemma is a version of Lemma 5.1 in [1] that works over non-binary alphabets.

**Lemma A.2.** *Fix a set of nodes $S$. Fix a node $u$ and a set of nodes $I$ that are not contained in $S$. Then*

$$\sqrt{\frac{1}{2}I(X_u; X_I|X_S)} \geq \nu_{u,I|S}$$

*Proof.*

$$\sqrt{\frac{1}{2}I(X_u; X_I|X_S)} = \sqrt{\frac{1}{2}\mathop{\mathbf{E}}_{X_S=x_S}[I(X_u; X_I|X_S = x_S)]}$$

$$\geq \mathop{\mathbf{E}}_{X_S=x_S}\left[\sqrt{\frac{1}{2}I(X_u; X_I|X_S = x_S)}\right]$$

$$= \mathop{\mathbf{E}}_{X_S=x_S}\left[\sqrt{\frac{1}{2}D_{KL}(\mathbf{Pr}(X_u, X_I|X_S = x_S)||\,\mathbf{Pr}(X_u|X_S = x_S)\,\mathbf{Pr}(X_I|X_S = x_S))}\right]$$

$$\geq \mathop{\mathbf{E}}_{X_S}\left[\sup_{R,G}[|\,\mathbf{Pr}(X_u = R, X_I = G|X_S) - \mathbf{Pr}(X_u = R|X_S)\,\mathbf{Pr}(X_I = G|X_S)|]\right]$$

$$\geq \mathop{\mathbf{E}}_{X_S}\left[\mathop{\mathbf{E}}_{R,G}[|\,\mathbf{Pr}(X_u = R, X_I = G|X_S) - \mathbf{Pr}(X_u = R|X_S)\,\mathbf{Pr}(X_I = G|X_S)|]\right]$$

$$= \nu_{u,I|S}.$$

where the first inequality follows from Jensen's inequality, and the second inequality follows from Pinsker's inequality. □

## A.3  No Cancellation

In this subsection we will show that a clique interaction of order $s$ cannot be completely cancelled out by clique interactions of lower order.

**Lemma A.3.** *Let $T^{1\cdots s}$ be a centered tensor of dimensions $d_1 \times \cdots \times d_s$ and suppose there exists at least one entry of $T^{1\cdots s}$ which is lower bounded in absolute value by a constant $\kappa$. For any $\ell < s$ and $i_1 < \cdots < i_\ell$ let $T^{i_1\cdots i_\ell}$ be an arbitrary centered tensor of dimensions $d_{i_1} \times \cdots \times d_{i_\ell}$. Define*

$$T(a_1,\ldots,a_s) = \sum_{\ell=1}^{s} \sum_{i_1 < \cdots < i_\ell} T^{i_1\cdots i_\ell}(a_{i_1},\ldots,a_{i_\ell}) \tag{7}$$

*and suppose the entries of $T$ are bounded by a constant $\mu$. Then for any $\ell$ and $i_1 < \cdots < i_\ell$, the entries of $T^{i_1\cdots i_\ell}(a_{i_1},\ldots,a_{i_\ell})$ are bounded above by $\mu\ell^\ell$.*

*Proof.* The sum over all values of indices $a_1,\ldots,a_s$ on the right hand side is zero, so the same must hold for the left hand side. Assume for contradiction that every entry of $T$ is upper bounded by $\mu$, to be optimized later. For each $m$ from 1 to $s$, consider summing over all of the indices except $a_m$, which is held fixed. Using that the sum over tensor fibers is zero, we observe that the right hand side of (8) is just

$$T^{i_m}(a_1) \prod_{m' \neq m} d_{m'}$$

and the left hand side is strictly bounded in norm by $\mu \prod_{m' \neq m} d_{m'}$ so $|T^{i_m}(a_m)| < \mu$ for all $a_m$. We have proven this for all $m$ from 1 to $s$.

Now we proceed by induction, assuming that $t$ indices are fixed. We will show that the entries of the $t$-tensors are bounded above by $\mu g(t)$ for $g(t) = 2^{t(t+1)/2}$ and have already proven this for $t = 1$. Now suppose we fix $a_1,\ldots,a_t$. We rearrange (8) to get

$$T(a_1,\ldots,a_s) - \sum_{\ell=1}^{t-1} \sum_{\{i_1<\cdots<i_\ell\} \subset [t]} T^{i_1\cdots i_\ell}(a_{i_1},\ldots,a_{i_\ell})$$

$$= T^{1\cdots t}(a_1,\ldots,a_t) + \sum_{\ell=1}^{s} \sum_{\{i_1<\cdots<i_\ell\} \not\subset [t]} T^{i_1\cdots i_\ell}(a_{i_1},\ldots,a_{i_\ell})$$

When we fix indices $a_1,\ldots,a_t$ and sum over the others, all but the first term on the rhs vanishes, and by applying the triangle inequality on the lhs and the induction hypothesis we get that

$$d_{t+1}\cdots d_s \left( \mu + \sum_{\ell=1}^{t-1} \binom{t}{\ell} \mu g(\ell) \right) > d_{t+1}\cdots d_s T^{ui_2\cdots i_t}(a_1,\ldots,a_t)$$

so taking $g(t)$ such that $g(0) = 1$ and

$$g(t) \geq \sum_{\ell=0}^{t-1} \binom{t}{\ell} g(\ell)$$

and in particular $g(t) = t^t$ works, because

$$t^t = (1 + (t-1))^t = \sum_{\ell=0}^{t} \binom{t}{\ell}(t-1)^\ell \geq \sum_{\ell=0}^{t-1} \binom{t}{\ell}\ell^\ell.$$

Thus we get that all the entries of $T^{i_1\cdots i_\ell}(a_{i_1},\ldots,a_{i_\ell})$ are bounded above by $\mu\ell^\ell$, which completes the proof. $\quad\square$

We are now ready to restate the above result in a more usable form:

**Lemma A.4.** *Let $T^{1\cdots s}$ be a centered tensor of dimensions $d_1 \times \cdots \times d_s$ and suppose there exists at least one entry of $T^{1\cdots s}$ which is lower bounded in absolute value by a constant $\kappa$. For any $\ell < s$ and $i_1 < \cdots < i_\ell$ let $T^{i_1 \cdots i_\ell}$ be an arbitrary centered tensor of dimensions $d_{i_1} \times \cdots \times d_{i_\ell}$. Let*

$$T(a_1, \ldots, a_s) = \sum_{\ell=1}^{s} \sum_{i_1 < \cdots < i_\ell} T^{i_1 \cdots i_\ell}(a_{i_1}, \ldots, a_{i_\ell}) \tag{8}$$

*Then the sum over all the entries of $T$ is 0, and there exists an entry of $T$ of absolute value lower bounded by $\kappa/s^s$.*

*Proof.* We apply the previous lemma with $\mu = \kappa/s^s$, and get that all the entries of $T^{1\cdots s}$ are bounded in absolute value by $\mu s^s$, giving a contradiction. □

# B  Omitted Proofs in Section 3

**Lemma B.1.**

$$\mathop{\mathbf{E}}_{Y,Z} \left[ \sum_R \sum_{B \neq R} \left( \mathcal{E}_{u,R}^Y - \mathcal{E}_{u,B}^Y - \mathcal{E}_{u,R}^Z + \mathcal{E}_{u,B}^Z \right) \left( \exp(\mathcal{E}_{u,R}^Y + \mathcal{E}_{u,B}^Z) - \exp(\mathcal{E}_{u,B}^Y + \mathcal{E}_{u,R}^Z) \right) \right] \geq \frac{4\alpha^2 \delta^{r-1}}{r^{2r} e^{2\gamma}}$$

*Proof.* Setting $a = \mathcal{E}_{u,R}^Y + \mathcal{E}_{u,B}^Z$ and $b = \mathcal{E}_{u,B}^Y + \mathcal{E}_{u,R}^Z$, letting $D' = K^3 \exp(2\gamma) \geq D$, and taking an expectation over the randomness in $Y$ and $Z$, we have

$$\mathop{\mathbf{E}}_{Y,Z} \left[ \sum_R \sum_{R \neq B} (a-b)(e^a - e^b) \right] = \mathbf{E}[\sum_R \sum_{R \neq B} (a-b) \int_b^a e^x dx]$$

$$\geq \mathbf{E}[\sum_R \sum_{R \neq B} (a-b)^2 e^{-2\gamma}] \geq \frac{1}{e^{2\gamma}} \sum_R \sum_{R \neq B} \mathbf{Var}[a-b]$$

where the inequality follows from the fact that $a, b \geq -2\gamma$. In the following claim, we give a more convenient expression for the above quantity.

**Claim B.2.**
$$\sum_R \sum_{R \neq B} \mathbf{Var}[a-b] = 4k_u \sum_R \mathbf{Var}[\mathcal{E}_{u,R}^Y]$$

*Proof.* Using the fact that $a - b = (\mathcal{E}_{u,R}^Y - \mathcal{E}_{u,B}^Y) + (\mathcal{E}_{u,B}^Z - \mathcal{E}_{u,R}^Z)$ we have that

$$\sum_R \sum_{R \neq B} \mathbf{Var}[a-b] = \sum_R \sum_{B \neq R} \mathbf{Var}[(\mathcal{E}_{u,R}^Y - \mathcal{E}_{u,B}^Y) + (\mathcal{E}_{u,B}^Z - \mathcal{E}_{u,R}^Z)]$$

$$= 2 \sum_R \sum_{B \neq R} \mathbf{Var}[(\mathcal{E}_{u,R}^Y - \mathcal{E}_{u,B}^Y)]$$

$$= 2 \sum_R \sum_{B \neq R} \left( 2\mathbf{Var}[\mathcal{E}_{u,R}^Y] - 2\mathbf{Cov}\left(\mathcal{E}_{u,R}^Y, \mathcal{E}_{u,B}^Y\right) \right)$$

$$= 2 \sum_R \left( 2(k_u - 1)\mathbf{Var}[\mathcal{E}_{u,R}^Y] - 2\mathbf{Cov}\left(\mathcal{E}_{u,R}^Y, \sum_{B \neq R} \mathcal{E}_{u,B}^Y\right) \right)$$

$$= 2 \sum_R \left( 2(k_u - 1)\mathbf{Var}[\mathcal{E}_{u,R}^Y] - 2\mathbf{Cov}\left(\mathcal{E}_{u,R}^Y, -\mathcal{E}_{u,R}^Y\right) \right)$$

$$= 4k_u \sum_R \mathbf{Var}[\mathcal{E}_{u,R}^Y]$$

where the second to last equality follows from the fact that the tensors are centered which gives $\sum_R \mathcal{E}_{u,R}^Y = 0$ for any $Y$. This completes the proof. □

Now we can complete the proof by appealing to the law of total variance. By assumption there is a maximal hyperedge $J = \{u, j_1 \ldots j_s\}$ containing $u$ with $|J| \leq r$, such that $\theta^{uJ}$ is $\alpha$-nonvanishing. Then we have

$$\sum_R \mathbf{Var}[\mathcal{E}_{u,R}^Y] \geq \sum_R \mathbf{Var}[\mathcal{E}_{u,R}^Y|Y_{\sim J}] = \sum_R \mathbf{Var}[T(R, Y_{j_1}, \ldots, Y_{j_s})|Y_{\sim J}]$$

where the tensor $T$ is defined by treating $Y_{\sim J}$ as fixed as follows

$$T(R, Y_{j_1}, \ldots, Y_{j_s}) = \sum_{\ell=2}^{r} \sum_{i_2 < \cdots < i_\ell} \theta^{u i_2 \cdots i_\ell}(R, Y_{i_2}, \cdots, Y_{i_\ell})$$

Now we claim there is a choice of $R$, $G$ and $G'$ so that $|T(R, G) - T(R, G')| > \alpha/r^r$. This follows because from Lemma A.4 we have that $T$ is $\alpha/r^r$-nonvanishing. Hence there is a choice of $R$ and $G$ so that $|T(R, G)| > \alpha/r^r$. Because $T$ is centered there must be a $G'$ so that $T(R, G')$ has the opposite sign.

Finally for this choice of $R$ we have

$$\mathbf{Var}[T(R, Y_{j_1}, \ldots, Y_{j_s})|Y_{\sim J}] \geq \frac{\alpha^2 \delta^{r-1}}{2r^{2r}}$$

which follows from the fact that $\mathbf{Pr}(Y_{J\setminus u} = G)$ and $\mathbf{Pr}(Y_{J\setminus u} = G')$ are both lower bounded by $\delta^{r-1}$ and the following elementary lower bound on the variance:

**Claim B.3.** *Let $Z$ be a random variable such that $Pr(Z = a) \geq p$ and $Pr(Z = b) \geq p$, then*

$$\mathbf{Var}(Z) \geq \frac{p}{2}(a - b)^2$$

*Proof.*

$$\mathbf{Var}(Z) \geq p(a - \mathbf{E}[Z])^2 + p(\mathbf{E}[Z] - b)^2 \geq p\left(a - \frac{a+b}{2}\right)^2 + p\left(b - \frac{a+b}{2}\right)^2 = \frac{p}{2}(a-b)^2$$

$\square$

Putting this all together we have

$$\mathop{\mathbf{E}}_{Y,Z}\left[\sum_R \sum_{R \neq B}(a - b)(e^a - e^b)\right] \geq \frac{4\alpha^2 \delta^{r-1}}{r^{2r} e^{2\gamma}}$$

which is the desired bound. This completes the proof. $\square$

## C Omitted Proofs in Section 4

### C.1 Mutual Information in Markov Random Fields

**Lemma C.1.** *For any strategy,*

$$\mathbf{E}[\Delta] \leq \gamma K \binom{D}{r-1} \mathop{\mathbf{E}}_{I, X_I, R}\left[\left|\mathbf{Pr}[X_u = R|X_I] - \mathbf{Pr}[X_u = R]\right|\right]$$

*Proof.* Intuitively this follows because Bob's optimal strategy given $I$, $X_I$ and $R$ is to guess
$$r = \text{sgn}(\mathbf{Pr}[X_u = R|X_I] - \mathbf{Pr}[X_u = R])\gamma K$$
More precisely, we have

$$\mathbf{E}[\Delta] = \mathop{\mathbf{E}}_{I, X_I, R}\left[\mathop{\mathbf{E}}_{X_{\sim I}, X'}\left[r\mathbb{1}_{X_u = R} - r\mathbb{1}_{X'_u = R}\Big|I, X_I, R\right]\right]$$

$$= \mathop{\mathbf{E}}_{I, X_I, R}\left[r\,\mathbf{Pr}[X_u = R|X_I] - r\,\mathbf{Pr}[X'_u = R]\right]$$

$$= \mathop{\mathbf{E}}_{I, X_I, R}\left[r\,\mathbf{Pr}[X_u = R|X_I] - r\,\mathbf{Pr}[X_u = R]\right]$$

$$\leq \gamma K \binom{D}{r-1} \mathop{\mathbf{E}}_{I, X_I, R}\left[\left|\mathbf{Pr}[X_u = R|X_I] - \mathbf{Pr}[X_u = R]\right|\right]$$

which completes the proof. $\square$

Next we lower bound the mutual information using (essentially) the same quantity. We prove
**Lemma C.2.**

$$\sqrt{\frac{1}{2}I(X_u; X_I)} \geq \frac{1}{K^r} \mathop{\mathbf{E}}_{X_I, R} \left[ |\mathbf{Pr}(X_u = R|X_I) - \mathbf{Pr}(X_u = R)| \right]$$

*Proof.* Applying Lemma A.2 with $S = \emptyset$ we have that

$$\sqrt{\frac{1}{2}I(X_u; X_I)} \geq \mathop{\mathbf{E}}_{R,G} \left[ |\mathbf{Pr}(X_u = R, X_I = G) - \mathbf{Pr}(X_u = R)\mathbf{Pr}(X_I = G)| \right]$$

$$= \mathop{\mathbf{E}}_{R,G} \left[ \mathbf{Pr}(X_I = G) |\mathbf{Pr}(X_u = R|X_I = G) - \mathbf{Pr}(X_u = R)| \right]$$

$$= \frac{1}{\prod_{i \in I} k_i} \sum_G \mathbf{Pr}(X_I = G) \mathop{\mathbf{E}}_{R}[|\mathbf{Pr}(X_u = R|X_I = G) - \mathbf{Pr}(X_u = R)|]$$

$$\geq \frac{1}{K^r} \mathop{\mathbf{E}}_{R,X_I} \left[ |\mathbf{Pr}(X_u = R|X_I) - \mathbf{Pr}(X_u = R)| \right]$$

where $R$ and $G$ are uniform (as in the definition of $\nu_{u,I|S}$). $\qquad\square$

Appealing to Lemma C.1, Lemma C.2 and Theorem 3.2 we conclude:

**Theorem C.3.** *Fix a non-isolated vertex $u$ contained in at least one $\alpha$-nonvanishing maximal hyperedge. Then taking $I$ uniformly at random from the subsets of the neighbors of $u$ of size $s = \min(r - 1, deg(u))$,*

$$\mathop{\mathbf{E}}_{I} \left[ \sqrt{\frac{1}{2}I(X_u; X_I)} \right] \geq \mathop{\mathbf{E}}_{I}[\nu_{u,I|\emptyset}] \geq C(\gamma, K, \alpha)$$

*where explicitly*

$$C(\gamma, K, \alpha) := \frac{4\alpha^2 \delta^{r-1}}{r^{2r} K^{r+1} \binom{D}{r-1} \gamma e^{2\gamma}}$$

## C.2  Extensions to Conditional Mutual Information

In the previous subsection, we showed that $X_u$ and $X_I$ have positive mutual information. Here we show that the argument extends to conditional mutual information when we condition on $X_S$ for any set $S$ that does not contain all the neighbors of $u$. The main idea is to show that there is a setting of $X_S$ where the hyperedges do not completely cancel out each other in the new Markov random field we obtain by conditioning on $X_S$.

More precisely fix a set of nodes $S$ that does not contain all the neighbors of $u$ and let $I$ be chosen uniformly at random from the subsets of neighbors of $u$ of size $s = \min(r - 1, |\Gamma(u) \setminus S|)$. Then we have

$$\mathop{\mathbf{E}}_{I}[\sqrt{\frac{1}{2}I(X_u; X_I|X_S)}] = \mathop{\mathbf{E}}_{I}[\sqrt{\frac{1}{2}\mathop{\mathbf{E}}_{X_S = x_S}[I(X_u; X_I|X_S = x_S)]}]$$

$$\geq \mathop{\mathbf{E}}_{I,X_S = x_S} \left[ \sqrt{\frac{1}{2}I(X_u; X_I|X_S = x_S)} \right]$$

which follows from Jensen's inequality. Now conditioned on $X_S = x_S$ the resulting distribution is again a Markov random field and $\gamma$ does not increase.

**Definition C.4.** Let $E$ be the event that conditioned on $X_S = x_S$, node $u$ is contained in at least one $\alpha/r^r$-nonvanishing maximal hyperedge.

**Lemma C.5.** $\mathbf{Pr}(E) \geq \delta^d$

*Proof.* When we fix $X_S = x_S$ we obtain a new Markov random field where the underlying hypergraph is

$$\mathcal{H}' := ([n] \setminus S, H') \text{ where } H' = \{h \setminus S | h \in H)$$

For notational convenience let $\phi(h)$ be the image of a hyperedge $h$ in $\mathcal{H}$ in the new hypergraph $\mathcal{H}'$. What makes things complicated is that a hyperedge in $\mathcal{H}'$ can have numerous preimages. The crux of our argument is in how to select the right one to show is $\alpha/r^r$-nonvanishing. First we observe that $u$ is contained in at least one non-empty hyperedge in $\mathcal{H}'$. This is because by assumption $S$ does not contain all the neighbors of $u$. Hence there is some neighbor $v \notin S$. Since $v$ is a neighbor of $u$ it means that there is a hyperedge $h \in H$ that contains both $u$ and $v$. In particular $\phi(h)$ contains $u$ and is nonempty.

Now that we know $u$ is not isolated in $\mathcal{H}'$, let $h^*$ be a hyperedge in $\mathcal{H}$ that contains $u$ and where $\phi(h^*)$ is maximal. Now let $f_1, f_2, \ldots f_p$ be the preimages of $\phi(h^*)$ so that without loss of generality $f_1$ is maximal in $\mathcal{H}$. Now let $J = \cup_{i=1}^p f_i \setminus \{u\}$. In particular, $J$ is the set of neighbors of $u$ that are contained in at least one of $f_1, f_2, \ldots f_p$. Finally let $J_1 = J \cap S := \{i_1, i_2, \ldots i_s\}$ and let $J_2 = J \setminus S := \{i'_1, i'_2, \ldots i'_{s'}\}$. We can now define

$$T(R, a_1, \ldots, a_s, a'_1, \ldots, a'_{s'}) = \sum_{i=1}^p \theta^{f_i}$$

which is the clique potential we get on hyperedge $\phi(h^*)$ when we fix each index in $J_1 \subseteq S$ to their corresponding value.

Suppose for the purposes of contradiction that all the entries of $T$ are strictly bounded in absolute value by $\alpha/r^r$. Then applying Lemma A.3 in the contrapositive we see that the entries of $f_1$ are strictly bounded above in absolute value by $\alpha$, but $f_1$ is maximal and thus $\alpha$-nonvanishing, which yields a contradiction. Thus there is some setting $a_1^*, \ldots, a_s^*$ such that the tensor

$$T'(R, a'_1, \ldots, a'_{s'}) = T(R, a_1^*, \ldots, a_s^*, a'_1, \ldots, a'_{s'})$$

has at least one entry with absolute value at least $\alpha/r^r$. Under this setting, $\phi(h^*)$ is $\alpha/r^r$-nonvanishing and by construction maximal in $\mathcal{H}'$ and thus we would be done. All that remains is to lower bound the probability of this setting. Since $J_1$ is a subset of the neighbors of $u$ we have $|J_1| \leq d$. Thus the probability that $(X_{i_1}, \ldots, X_{i_s}) = (a_1^*, \ldots, a_s^*)$ is bounded below by $\delta^s \geq \delta^d$, which completes the proof. $\qquad\square$

Now we are ready to prove a lower bound on conditional mutual information:

**Theorem C.6.** *Fix a vertex $u$ such that all of the maximal hyperedges containing $u$ are $\alpha$-nonvanishing, and a subset of the vertices $S$ which does not contain the entire neighborhood of $u$. Then taking $I$ uniformly at random from the subsets of the neighbors of $u$ not contained in $S$ of size $s = \min(r - 1, |\Gamma(u) \setminus S|)$,*

$$\mathbf{E}_I\left[\sqrt{\frac{1}{2}I(X_u; X_I|X_S)}\right] \geq E_I[\nu_{u,I|S}] \geq C'(\gamma, K, \alpha)$$

*where explicitly*

$$C'(\gamma, K, \alpha) := \frac{4\alpha^2 \delta^{r+d-1}}{r^{2r} K^{r+1} \binom{D}{r-1} \gamma e^{2\gamma}}$$

*Proof.* We have

$$\mathbf{E}_{I,X_S}\left[\sqrt{\frac{1}{2}I(X_u; X_I|X_S)}\right] \geq \mathbf{E}_{I,X_S=x_S}\left[\sqrt{\frac{1}{2}I(X_u; X_I|X_S = x_S)}\mathbb{1}_E\right] \geq \delta^d C(\gamma, K, \alpha)$$

where the last inequality follows by invoking Lemma C.5 and applying Theorem C.3 to the new Markov random field we get by conditioning on $X_S = x_S$. $\qquad\square$

# D   Omitted Proofs in Section 5

## D.1   Learning Markov Random Fields

The algorithm will succeed provided that $\widehat{\nu}_{u,I|S}$ is sufficiently close to the true value $\nu_{u,I|S}$. This motivates the definition of the event $A$:

**Definition D.1.** We denote by $A(\ell, \epsilon)$ the event that for all $u$, $I$ and $S$ with $|I| \leq r - 1$ and $|S| \leq \ell$ simultaneously,

$$\left| \nu_{u,i|S} - \widehat{\nu}_{u,i|S} \right| < \epsilon.$$

We let $A$ denote the event $A(L, \tau/2)$.

The proof of the following technical lemma is left to an appendix.

**Lemma D.2.** *Fix a set $S$ with $|S| \leq \ell$ and suppose that for any set $T \supseteq S$ with $|T \setminus S| \leq r$, that*

$$|\widehat{\mathbf{Pr}}(X_T = x_T) - \mathbf{Pr}(X_T = x_T)| < \sigma.$$

*If $\sigma \leq \epsilon K^{-\ell} \frac{\delta^\ell}{5}$ then for any $I$ with $|I| \leq r - 1$,*

$$\left| \nu_{u,i|S} - \widehat{\nu}_{u,i|S} \right| < \epsilon.$$

**Lemma D.3.** *Fix $\ell, \epsilon$ and $\omega > 0$. If the number of samples satisfies*

$$m \geq \frac{15 K^{2\ell}}{\epsilon^2 \delta^{2\ell}} \Big( \log(1/\omega) + \log(\ell + r) + (\ell + r) \log(nK) + \log 2 \Big)$$

*then $\mathbf{Pr}(A(\ell, \epsilon)) \geq 1 - \omega$.*

*Proof of Lemma D.3.* Fix $\ell, \epsilon$ and $\omega > 0$. Let $m$ denote the number of samples. By Hoeffding's inequality, for any set $T$,

$$\mathbf{Pr}[|\widehat{\mathbf{Pr}}(X_T = x_T) - \mathbf{Pr}(X_T = x_T)| > \sigma] \leq 2 \exp(-2\sigma^2 m)$$

and taking the union bound over all possibly $x_T$ for $T$ with $|T| \leq \ell + r$, of which there are at most

$$\sum_{i=1}^{\ell+r} \binom{n}{i} K^i \leq \sum_{i=1}^{\ell+r} (nK)^i \leq (\ell + r)(nK)^{\ell+r}$$

many, we find the probability that $|\widehat{\mathbf{Pr}}(X_T = x_T) - \mathbf{Pr}(X_T = x_T)| > \sigma$ for any such $T$ is at most

$$(\ell + r)(nK)^{\ell+r} 2 \exp(-2\sigma^2 m)$$

Therefore taking

$$m \geq \frac{\log(1/\omega) + \log(\ell + r) + (\ell + r)\log(nK) + \log 2}{2\sigma^2} \tag{9}$$

ensures this probability is at most $\omega$.

Now applying Lemma D.2 and substituting $\sigma = \epsilon K^{-\ell} \frac{\delta^\ell}{5}$ into (9), we see that the result holds if

$$m \geq \frac{15 K^{2\ell}}{\epsilon^2 \delta^{2\ell}} \Big( \log(1/\omega) + \log(\ell + r) + (\ell + r) \log(nK) + \log 2 \Big)$$

$\square$

**Lemma D.4.** *Assume that the event $A$ holds. Then every time a node $i$ is added to $S$ in Step 2 of the algorithm, the mutual information $I(X_u; X_S)$ increases by at least $\tau^2/8$.*

*Proof.* For a particular iteration of Step 2, let $I$ denote the newly added set of nodes, and $S$ the set of candidate neighbors before adding $I$. Then we must show for $Q = \tau^2/8$ that

$$I(X_u; X_{S \cup \{I\}}) \geq I(X_u; X_S) + Q$$

which by the chain rule for expectation is equivalent to

$$I(X_u; X_I | X_S) \geq Q.$$

Applying Lemma A.2 and the fact that event $A$ holds, we see

$$\sqrt{\frac{1}{2} \cdot I(X_u; X_I | X_S)} \geq \frac{1}{2} \nu_{u,I|S} \geq \frac{1}{2} \left( \widehat{\nu}_{u,i|S} - \tau/2 \right)$$

Thus the algorithm only adds node $i$ to $S$ if $\widehat{\nu}_{u,i|S} \geq \tau$, so the chain of inequalities implies that

$$I(X_u; X_i | X_S) \geq \frac{1}{2} (\tau - \tau/2)^2 = \tau^2/8$$

$\square$

**Lemma D.5.** *If event $A$ holds then at the end of Step 2, $S$ contains all of the neighbors of $u$.*

*Proof.* Step 2 ended either because $|S| > L$ or because there was no set of nodes $I \subset S^C$ with $\widehat{\nu}_{u,I|S} > \tau$. First we rule out the former possibility. Whenever a new element is added to $S$, the quantity $I(X_u; X_S)$ increases by at least $\tau^2/8$. But

$$I(X_u; X_S) \leq H(X_u) \leq \log K$$

because $X_u$ takes on at most $K$ states. Thus if $|S| > L$ then

$$\log K \geq I(X_u; X_S) > L(\tau^2/8) = \log K$$

which gives a contradiction.

Thus at the end of Step 2 we must have that there is no set of nodes $I \subset S^C$ with $\widehat{\nu}_{u,I|S} > \tau$. Suppose for the purposes of contradiction that $S$ does not contain all of the neighbors of $u$. Then by Theorem C.6, there exists a subset of the neighbors such that $\nu_{u,I|S} \geq C'(\gamma, k, \alpha) = 2\tau$, and because event $A$ holds we know $\widehat{\nu}_{u,I|S} > 2\tau - \tau/2 > \tau$, which gives us our contradiction and completes the proof of the lemma. $\qquad\square$

**Lemma D.6.** *If event $A$ holds and if at the start of Step 3 $S$ contains all neighbors of $u$, then at the end of Step 3 the remaining set of nodes are exactly the neighbors of $u$.*

*Proof.* If $A(\ell)$ holds, then during Step 3,

$$\widehat{\nu}_{u,i|S\setminus\{i\}} < \nu_{u,i|S} + \tau/2 \leq \sqrt{\frac{1}{2}I(X_u; X_i|X_S)} + \tau/2 = \tau/2$$

for all nodes $i$ that are not neighbors of $u$. Thus all such nodes are pruned. Furthermore, by Theorem C.6, $\widehat{\nu}_{u,i|S\setminus\{i\}} > \nu_{u,i|S\setminus\{i\}} - \tau/2 \geq 2\tau - \tau/2 = 3\tau/2$ for all neighbors of $u$ and thus no neighbor is pruned. This completes the proof. $\qquad\square$

Recall that $\gamma \leq \beta r D^r$, $\delta = e^{-2\gamma}/K$, $(C'(\gamma, K, \alpha))^{-1} = O(\frac{K^{r+1}r^{2r}}{\alpha^2\delta^{2D}}D^{r-1}\gamma e^{-2\gamma})$ and $L = O(C'(\gamma, K, \alpha)^{-2})$.

**Theorem D.7.** *Fix $\omega > 0$. Suppose we are given $m$ samples from an $\alpha, \beta$-non-degenerate Markov random field with $r$-order interactions where the underlying graph has maximum degree at most $D$ and each node takes on at most $K$ states. Suppose that*

$$m \geq \frac{60K^{2L}}{\tau^2\delta^{2L}}\Big(\log(1/\omega) + \log(L + r) + (L + r)\log(nK) + \log 2\Big).$$

*Then with probability at least $1 - \omega$, MRFNBHD when run starting from each node $u$ recovers the correct neighborhood of $u$, and thus recovers the underlying graph $G$. Furthermore, each run of the algorithm takes $O(mLn^r)$ time.*

*Proof.* Set $\ell = L$ and $\epsilon = \tau/2$ in Lemma D.3. Then event $A$ occurs with probability at least $1 - \omega$ for our choice of $m$. Now by Lemma D.5 and Lemma D.6 the algorithm returns the correct set of neighbors of $u$ for every node $u$.

To analyze the running time, observe that when running algorithm MRFNBHD at a single node $u$, the bottleneck is Step 2, in which there are at most $L$ steps and in each step the algorithm must loop over all subsets of the vertices in $[n] \setminus S$ of size $r - 1$, of which there are $\sum_{\ell=1}^{r-1}\binom{n}{\ell} = O(n^{r-1})$ many. Running the algorithm at all nodes thus takes $O(mLn^r)$ time. $\qquad\square$

**Remark D.8.** Note that when we plug in the values of $\gamma$ and $\delta$ we get that the overall sample complexity of our algorithm in terms of $D$ and $r$ is doubly exponential in $D^r$.

## D.2 Extensions

### D.2.1 Learning with Bounded Queries

Our starting point is an elementary observation about MRFNBHD:

**Observation 1.** *In Step 2,* MRFNBHD *only needs* $\widehat{\nu}_{u,I|S}$ *for all $I$ with $|I| \leq r - 1$. Similarly at Step 3,* MRFNBHD *only needs* $\widehat{\nu}_{u,i|S\setminus i}$ *for each $i \in S$.*

Thus the number of distinct terms $\widehat{\nu}_{u,I|S}$ which MRFNBHD needs is at most $L(r-1)n^{r-1}$ for Step 2 and $R$ for Step 3, which in total is at most $Lrn^{r-1}$.

**Lemma D.9.** *Fix a node $u$, a set $S$ with $\ell = |S|$, a set $I$ with $|I| \leq r - 1$ and fix $\epsilon$ and $\omega > 0$. If the number of samples we observe of $X_{S\cup I\cup\{u\}}$ satisfies*

$$m' \geq \frac{15K^{2\ell}}{\epsilon^2\delta^{2\ell}}\Big(\log(1/\omega) + \log(\ell + r) + (\ell + r)\log(nK) + \log 2\Big)$$

*then*

$$|\nu_{u,I|S} - \widehat{\nu}_{u,I|S}| < \epsilon$$

*with probability at least $1 - \omega$.*

*Proof.* This follows by the same Hoeffding and union bound as in proof of Lemma D.3. $\qquad\square$

**Theorem D.10.** *Fix an $\alpha, \beta$-non-degenerate Markov random field with $r$-order interactions where the underlying graph has maximum degree at most $D$ and each node takes on at most $K$ states. The bounded queries modification to the algorithm returns the correct neighborhood of every vertex $u$ using $m'Lrn^r$ bounded queries of size at most $L + r$ where*

$$m' = \frac{60K^{2L}}{\tau^2\delta^{2L}}\Big(\log(Lrn^r/\omega) + \log(L + r) + (L + r)\log(nK) + \log 2\Big),$$

*with probability at least $1 - \omega$.*

*Proof.* Invoking Lemma D.9 with $\omega' = \frac{\omega}{Lrn^r}$, $\epsilon = \tau/2$ and $\ell = L$, we get that each query to $\widehat{\nu}_{u,I|S}$ fails (i.e. is wrong by at least $\tau/2$) with probability at most $\frac{\omega}{Lrn^r}$. We observed that Algorithm MRFNBHD makes at most $Lrn^{r-1}$ queries of the form, $\widehat{\nu}_{u,I|S}$. Therefore, by a union bound, with probability at least $1 - \omega/n$, the bounded queries algorithm answers all of those queries to within tolerance $\tau/2$.

Now it follows as in Theorem D.7 that the algorithm returns the correct neighborhood of node $u$ with probability at least $1-\omega/n$, and taking the union bound over all nodes $u$ it follows that the algorithm recovers the correct neighborhood of all nodes with probability at least $1 - \omega$. This completes the proof. $\qquad\square$

### D.2.2 Learning with Random Erasures

Here we consider another variant where we do not observe full samples from a Markov random field. Instead we observe partial samples where the state of each node is revealed independently with probability $p$, and the choice of which nodes to reveal is independent of the sample. We can apply our algorithm in this setting, as follows.

**Lemma D.11.** *With probability at least $1 - \epsilon$, if we take $N\frac{\ell\log n+\log\ell+\log N/\epsilon}{p^2}$ samples then we will see each set $S$ at least $N$ times for every $|S| \leq \ell$.*

*Proof.* Each sample has at least a $p^\ell$ chance of being observed, and there are at most $\ell n^\ell$ many different sets $S$. So by a union bound,

$$Pr[\text{exists unobserved } S \text{ after } t \text{ steps}] \leq n^\ell(1 - p^\ell)^t \leq \epsilon/N$$

if we take $t = \frac{\ell\log n+\log\ell+\log N/\epsilon}{p^2}$. Repeating this $N$ times, we see that with

$$Nt = N\frac{\ell\log n + \log\ell + \log N/\epsilon}{p^2}$$

many samples, we see every $S$ at least $N$ times with probability at least $1 - \epsilon$. $\qquad\square$

**Lemma D.12.** *Fix $\ell, \epsilon$ and $\omega > 0$. If the number of samples satisfies*

$$m \geq N \frac{\ell \log N + \log \ell + \log 2N/\omega}{p^2}$$

*where*

$$N = \frac{15 K^{2\ell}}{\epsilon^2 \delta^{2\ell}} \Big( \log(2/\omega) + \log(\ell + r) + (\ell + r) \log(nK) + \log 2 \Big)$$

*then $\mathbf{Pr}(A(\ell, \epsilon)) \geq 1 - \omega$.*

*Proof.* Observe by Lemma D.11, taking $\epsilon = \omega/2$ that with probability at least $1 - \omega/2$, for every set $S$ with $|S| \leq \ell$ we see at least $N$ samples revealing all of the members of $S$. Condition on this event; now the proof is exactly the same as Lemma D.3 taking $\omega' = \omega/2$, applying Hoeffding, Lemma D.2 and taking the union bound, we see that event $A$ holds with probability at least $\omega/2$. Therefore the total probability $A$ occurs is at least $1 - \omega/2 - \omega/2 = 1 - \omega$. $\qquad\square$

**Theorem D.13.** *Fix $\omega > 0$. Suppose we are given $m$ samples from an $\alpha, \beta$-non-degenerate Markov random field with $r$-order interactions where the underlying graph has maximum degree at most $D$ and each node takes on at most $K$ states. Suppose that*

$$m \geq N \frac{\ell \log n + \log L + \log 2N/\omega}{p^2}$$

*where*

$$N = \frac{60 K^{2L}}{\tau^2 \delta^{2L}} \Big( \log(2/\omega) + \log(L + r) + (L + r) \log(nK) + \log 2 \Big).$$

*Then with probability at least $1 - \omega$,* MRFNBHD *when run starting from each node $u$ recovers the correct neighborhood of $u$, and thus recovers the underlying graph $G$. Furthermore, each run of the algorithm takes $O(mLn^r)$ time.*

*Proof.* By Lemma D.12, given our assumption on $m$ the event $A$ occurs with probability at least $1 - \omega$. Conditioned on event $A$, the algorithm returns the correct answer by the same argument as Theorem D.7. $\qquad\square$

# E    Proof of Lemma D.2

*Proof.* Observe the left hand side of our desired inequality is bounded by

$$E_{R,G} \Big| \widehat{\mathbf{E}}_{X_S} [|\widehat{\mathbf{Pr}}(X_u = R, X_I = G|X_S) - \widehat{\mathbf{Pr}}(X_u = R|X_S)\widehat{\mathbf{Pr}}(X_I = G|X_S)|]$$
$$- \underset{X_S}{\mathbf{E}} [|\mathbf{Pr}(X_u = R, X_I = G|X_S) - \mathbf{Pr}(X_u = R|X_S)\mathbf{Pr}(X_I = G|X_S)|] \Big|$$

So it suffices if we can bound for every $R$ and $G$

$$\Big| \widehat{\mathbf{E}}_{X_S} [|\widehat{\mathbf{Pr}}(X_u = R, X_I = G|X_S) - \widehat{\mathbf{Pr}}(X_u = R|X_S)\widehat{\mathbf{Pr}}(X_I = G|X_S)|]$$
$$- \underset{X_S}{\mathbf{E}} [|\mathbf{Pr}(X_u = R, X_I = G|X_S) - \mathbf{Pr}(X_u = R|X_S)\mathbf{Pr}(X_I = G|X_S)|] \Big|$$
$$= \Big| \sum_{x_S} |\widehat{\mathbf{Pr}}(X_u = R, X_I = G, X_S = x_S) - \widehat{\mathbf{Pr}}(X_u = R|X_S = x_S)\widehat{\mathbf{Pr}}(X_I = G, X_S = x_S)|$$
$$- |\mathbf{Pr}(X_u = R, X_i = G, X_S = x_S) - \mathbf{Pr}(X_u = R|X_S = x_S)\mathbf{Pr}(X_I = G, X_S = x_S)| \Big|$$
$$\leq \sum_{x_S} \Big| |\widehat{\mathbf{Pr}}(X_u = R, X_I = G, X_S = x_S) - \widehat{\mathbf{Pr}}(X_u = R|X_S = x_S)\widehat{\mathbf{Pr}}(X_I = G, X_S = x_S)|$$
$$- |\mathbf{Pr}(X_u = R, X_I = G, X_S = x_S) - \mathbf{Pr}(X_u = R|X_S = x_S)\mathbf{Pr}(X_I = G, X_S = x_S)| \Big|$$
$$\leq \sum_{x_S} \Big| \widehat{\mathbf{Pr}}(X_u = R, X_I = G, X_S = x_S) - \widehat{\mathbf{Pr}}(X_u = R|X_S = x_S)\widehat{\mathbf{Pr}}(X_I = G, X_S = x_S)$$
$$- \mathbf{Pr}(X_u = R, X_I = G, X_S = x_S) + \mathbf{Pr}(X_u = R|X_S = x_S)\mathbf{Pr}(X_I = G, X_S = x_S) \Big|$$

$$\leq \sum_{x_S} |\widehat{\mathbf{Pr}}(X_u = R, X_I = G, X_S = x_S) - \mathbf{Pr}(X_u = R, X_I = G, X_S = x_S)|$$

$$+ \sum_{x_S} |\widehat{\mathbf{Pr}}(X_u = R | X_S = x_S)\widehat{\mathbf{Pr}}(X_I = G, X_S = x_S) - \mathbf{Pr}(X_u = R | X_S)\mathbf{Pr}(X_I = G, X_S = x_S)|$$

$$\leq K^{|S|}\sigma + \sum_{x_S} |\widehat{\mathbf{Pr}}(X_u = R | X_S = x_S)\widehat{\mathbf{Pr}}(X_I = G, X_S = x_S) - \mathbf{Pr}(X_u = R | X_S = x_S)\mathbf{Pr}(X_I = G, X_S = x_S)|$$

To bound the second term, observe

$$|\widehat{\mathbf{Pr}}(X_u = R | X_S = x_S)\widehat{\mathbf{Pr}}(X_I = G, X_S = x_S) - \mathbf{Pr}(X_u = R | X_S)\mathbf{Pr}(X_I = G, X_S = x_S)|$$

$$\leq \widehat{\mathbf{Pr}}(X_u = R | X_S = x_S)|\widehat{\mathbf{Pr}}(X_I = G, X_S = x_S) - \mathbf{Pr}(X_I = G, X_S = x_S)|$$

$$+ \mathbf{Pr}(X_I = G, X_S = x_S)|\widehat{\mathbf{Pr}}(X_u = R | X_S = x_S) - \mathbf{Pr}(X_u = R | X_S)|$$

$$\leq |\widehat{\mathbf{Pr}}(X_I = G, X_S = x_S) - \mathbf{Pr}(X_I = G, X_S = x_S)| + |\widehat{\mathbf{Pr}}(X_u = R | X_S = x_S) - \mathbf{Pr}(X_u = R | X_S)|$$

$$\leq \sigma + |\widehat{\mathbf{Pr}}(X_u = R | X_S = x_S) - \mathbf{Pr}(X_u = R | X_S)|$$

and furthermore

$$|\widehat{\mathbf{Pr}}(X_u = R | X_S = x_S) - \mathbf{Pr}(X_u = R | X_S = x_S)|$$

$$= \left| \frac{\widehat{\mathbf{Pr}}(X_u = R, X_S = x_S)}{\widehat{\mathbf{Pr}}(X_S = x_S)} - \frac{\mathbf{Pr}(X_u = R, X_S = x_S)}{\mathbf{Pr}(X_S = x_S)} \right|$$

$$\leq \left| \frac{\widehat{\mathbf{Pr}}(X_u = R, X_S = x_S)}{\widehat{\mathbf{Pr}}(X_S = x_S)} - \frac{\mathbf{Pr}(X_u = R, X_S = x_S)}{\widehat{\mathbf{Pr}}(X_S = x_S)} \right|$$

$$+ \left| \frac{\mathbf{Pr}(X_u = R, X_S = x_S)}{\widehat{\mathbf{Pr}}(X_S = x_S)} - \frac{\mathbf{Pr}(X_u = R, X_S = x_S)}{\mathbf{Pr}(X_S = x_S)} \right|$$

$$\leq \frac{\sigma}{\delta^{|S|}} + \mathbf{Pr}(X_u = R, X_S = x_S) \left| \frac{\mathbf{Pr}(X_S = x_S) - \widehat{\mathbf{Pr}}(X_S = x_S)}{\widehat{\mathbf{Pr}}(X_S = x_S)\mathbf{Pr}(X_S = x_S)} \right| \leq \frac{\sigma}{\delta^{|S|}} + \frac{\sigma}{\delta^{|S|} - \sigma}$$

Finally, if $\sigma < \epsilon K^{-\ell}\frac{\delta^{\ell}}{5}$ then because $|S| \leq \ell$ and $\sigma < \delta^{\ell}/5 < \delta^{\ell}/2$

$$K^{|S|}\sigma + \sum_{x_S} \left( \sigma + \frac{\sigma}{\delta^{|S|}} + \frac{\sigma}{\delta^{|S|} - \sigma} \right) = K^{|S|}\sigma \left( 2 + \frac{1}{\delta^{|S|}} + \frac{1}{\delta^{|S|} - \sigma} \right)$$

$$< K^{|S|}\sigma \left( \frac{2}{\delta^{|S|}} + \frac{1}{\delta^{|S|}} + \frac{2}{\delta^{|S|}} \right) < \epsilon$$

$\square$

## Footnotes

[1] We treat $C$ as the lowest order interaction, so when we are subtracting from the 1-tensors (vectors) $\theta^i$ to recenter them, we add the corresponding amount to $C$.

## References

[1] Guy Bresler. Efficiently learning ising models on arbitrary graphs. In *Proceedings of the Forty-Seventh Annual ACM on Symposium on Theory of Computing*, pages 771–782. ACM, 2015.