[Reviews · NeurIPS 2017]

Reviewer 1



This paper is concerned with learning Markov random fields (MRF). It is a theoretical paper, which is ultimately focused with proving a particular statement: given a variable X in an MRF, and given some of its Markov blanket variables B, there exists another variable Y that is conditionally dependent on X given the subset of B. In general this statement is not true; so the goal here is to identify some conditions where this is true. Most of this paper is centered around this, from which the ability to learn an MRF follows. The paper is mostly technical; my main complaint is that I do not think it is very intuitive. It appears that central to the results is the assumption on non-degeneracy, which I believe should be explained in higher level terms. Intuitively, there are a number of situations where the above statement would not hold. For example, if a potential is allowed to have determinism, which would lead to the MRF having inputs with zero probability, then one could construct a potential that makes some variable functionally dependent on other variables. This would allow one to construct an MRF where conditioning on some members of a Markov blanket would fix the values of other members, hence separating a variable from the rest of the network. However, this appears to be ruled out by the non-degeneracy conditions. Another simple condition is if a potential were uniform. In this case, some edges on the MRF could be vacuous and could be removed. This also appears to be ruled out by the non-degeneracy conditions. A more general condition is if a potential over multiple variables actually represented (for example) the product of unary potentials. Again, this would involve vacuous MRF edges, although it is not immediately clear to me if this case is ruled out as well. The point however is that it would be helpful for me to know what these non-degeneracy conditions imply, at a more abstract level. For example, do they rule out the above cases? Are there other cases that need to be ruled out? This would help in absorbing the technicalities. Another place where this would help is in justifying the machinery needed to prove the desired result. As the authors state, the result is "simultaneously surprising and not surprising." Going further, is there a set of simple and intuitive conditions (but possibly stronger conditions) that would lead to analogous results on learning MRFs? This would also be helpful in making the paper clearer.

Reviewer 2



The authors consider the problem of automatically learning the structure of a general Markov Random Field model. Their main contribution is a lower bound on the mutual information between a variable u and a variable v conditioned on any set of variables which does not contain the full Markov blanket of u. The bound depends on the maximum degree of the MRF graph, the size of the variable state spaces, the order of the interactions and some broad conditions on the value of the clique potentials. From this result, the authors derive an algorithm which can recover the structure of an MRF with high probability in "polynomial" time (exponential in the order of the potentials), and with bounded queries. This paper is a bit of an outlier for NIPS, which isn't necessarily a bad thing, but should come with some form of disclaimer. Both the bound and algorithms have theoretical significance. However, it should be clearly stated that the proposed algorithms are given as a proof of existence, rather than as a method to follow: the form of the constants make any actual guaranteed application intractable, to say nothing of the sample complexity to have sufficiently accurate estimates of the mutual information values. If the authors believe that the algorithms could still yield reasonable results, then this should be demonstrated at the very least on artificial data.

Reviewer 3



The setup in Eq. 1 is unclear on a first reading. What is the exact meaning of r here? This is a critical constant in the major result proven, and not carefully defined anywhere in the paper. I *think* that r is the maximum clique size. This is particularly ambiguous because of the statement that theta^{i1,i2,...,il}(a1,a2,...,an) "is assumed to be zero on non-cliques". I've assumed the meaning of this is that, more precisely, that function ignores aj unless there is some value t such that aj=it. Then, this makes sense as a definition of a Markov random field with maximum clique size r. (I encourage clarification of this in the paper if correct, and a rebuttal if I have misunderstood this.) In any case, this is an unusual definition of an MRF in the NIPS community. I think the paper would have more impact if a more standard definition of an MRF were used in the introduction, and then the equivalence/conversion were just done in the technical part. The main result only references the constant r, so no need to introduce all this just to state the result. On a related note, it's worth giving a couple sentences on the relationship to Bresler's result. (For an Ising model, r=2, etc.) In particular, one part of this relationship is a bit unclear to me. As far as I know, Bresler's result applies to arbitrary Ising models, irrespective of graph structure. So, take an Ising model with a dense structure so that there are large cliques. In this case, it seems to me that the current paper's result will use (r-1) nodes while Bresler's will still use a single node-- thus in that particular case the current paper's result is weaker. Is this correct? (It's not a huge weakness even if true but worth discussing in detail since it could guide future work.) [EDIT: I see now that this is *not* the case, see response below] I also feel that the discussion of Chow-Liu is missing a very important aspect. Chow-Liu doesn't just correctly recover the true structure when run on data generated from a tree. Rather, Chow-Lui finds the *maximum-likelihood* tree for data from an *arbitrary* distribution. This is a property that almost all follow-up work does not satisfy (Srebro showed bounds). The discussion in this paper is all true, but doesn't mention that "maximum likelihood" or "model robustness" issue at all, which is hugely important in practice. For reference: The basic result is that given a single node $u$, and a hypothesized set of "separators" $S$ (neighbors of $u$) then there will be some set of nodes $I$ with size at most $r-1$ such that $u$ and $I$ have positive conditional mutual information. The proof of the central result proceeds by setting up a "game", which works as follows: 1) We pick a node $X_u$ to look at. 2) Alice draws two joint samples $X$ and $X'$. 3) Alice draws a random value $R$ (uniformly from the space of possible values of $X_u$ 4) Alice picks a random set of neighbors of $X_u$, call them $X_I$. 5) Alice tells Bob the values of $X_I$ 6) Bob get's to wager on if $X_u=R$ or $X'_u=R$. Bob wins his wager if $X_u=R$ an loses his wager if $X'_u=R$ and nothing happens if both or neither are true. Here I first felt like I *must* be missing something, since this is just establishing that $X_u$ has mutual information with it's neighbors. (There is no reference to the "separator" set S in the main result.) However, it later appears that this is just a warmup (regular mutual information) and can be extended to the conditional setting. Actually, couldn't the conditional setting itself be phrased as a game, something like 1) We pick a node $X_u$ and a set of hypothesized "separators" $X_S$ to look at. 2) Alice draws two joint samples $X$ and $X'$ Both are conditioned on the some random value for $X_S$. 3) Alice draws a random value $R$ (uniformly from the space of possible values of $X_u$ 4) Alice picks a random set of nodes (not including $X_u$ or $X_S$, call them $X_I$. 5) Alice tells Bob the values of $X_I$ 6) Bob get's to wager on if $X_u=R$ or $X'_u=R$. Bob wins his wager if $X_u=R$ an loses his wager if $X'_u=R$ and nothing happens if both or neither are true. I don't think this adds anything to the final result, but is an intuition for something closer to the final goal. After all this, the paper discusses an algorithm for greedily learning an MRF graph (in sort of the obvious way, by exploiting the above result) There is some analysis of how often you might go wrong estimating mutual information from samples, which I appreciate. Overall, as far as I can see, the result appears to be true. However, I'm not sure that the purely theoretical result is sufficiently interesting (at NIPS) to be published with no experiments. As I mentioned above, Chow-Liu has the major advantage of finding the maximum likelihood solution, which the current method does not appear to have. (It would violate hardness results due to Srebro.) Further, note that the bound given in Theorem 5.1, despite the high order, is only for correctly recovering the structure of a single node, so there would need to be another lever of applying the union bound to this result with lower delta to get a correct full model. EDIT AFTER REBUTTAL: Thanks for the rebuttal. I see now that I should understand $r$ not as the maximum clique size, but rather as the maximum order of interactions. (E.g. if one has a fully-connected Ising model, you would have r=2 but the maximum clique size would be n). This answers the question I had about this being a generalization of Bressler's result. (That is, this paper's is a strict generalization.) This does slightly improve my estimation of this paper, though I thought this was a relatively small concern in any case. My more serious concerns are that a pure theoretical result is appropriate for NIPS.